# Epidemiological Overview of Overweight and Obesity Related to Eating Habits, Physical Activity and the Concurrent Presence of Depression and Anxiety in Adolescents from High Schools in Mexico City: A Cross-Sectional Study

**DOI:** 10.3390/healthcare12060604

**Published:** 2024-03-07

**Authors:** Rey Gutiérrez Tolentino, Irina Lazarevich, Manuel Abraham Gómez Martínez, Jorge Armando Barriguete Meléndez, Beatriz Schettino Bermúdez, José Jesús Pérez González, Rubén del Muro Delgado, Claudia Cecilia Radilla Vázquez

**Affiliations:** 1Division of Biological and Health Sciences, Universidad Autónoma Metropolitana-Xochimilco, Mexico City 04960, Mexico; reygut@correo.xoc.uam.mx (R.G.T.); iboris@correo.xoc.uam.mx (I.L.); schettin@correo.xoc.uam.mx (B.S.B.); jjperez@correo.xoc.uam.mx (J.J.P.G.); delmuro@correo.xoc.uam.mx (R.d.M.D.); 2Independent Researcher, Mexico City 14080, Mexico; gommar1103@gmail.com; 3Nutrition Division & Metabolism Lab, Anahuac University, Mexico City 52786, Mexico; consejodesalud@edomex.gob.mx; 4Fundación Aprende con Reyhan A.C., Mexico City 04640, Mexico

**Keywords:** overweight, obesity, adolescents, habits, depression, anxiety

## Abstract

Overweight and obesity in adolescents has become a serious public health problem worldwide and Mexico City is no exception. Therefore, the objective of this study was to investigate the epidemiological panorama of overweight and obesity related to eating habits, physical activity and the concurrent presence of depression and anxiety in adolescents from high schools in Mexico City. Anthropometric measurements were taken from 2710 adolescents from 33 participating high schools. Likewise, a previously validated eating habit and physical activity questionnaire was administered, which consisted of four different sections, where each of the sections focused on key aspects of the participants’ lifestyle: (1) eating habits, (2) intake of non-recommended foods, (3) food and company environment, and (4) physical activity. Moreover, the Hospital Anxiety and Depression Scale (HADS) for anxiety and depression was applied. In this study, a high prevalence of overweight and obesity (26.5% overweight and 20.0% obese) was found in adolescents from high schools in Mexico City. Only 13.14% of participants had adequate eating habits and 18.19% physical activity habits. An association was found between having inadequate eating habits and obesity in adolescent women (OR = 1.95; CI 1.009–3.76). Additionally, associations were observed between depression symptoms and obesity (OR = 5.68, CI 1.36–32.81; *p* = 0.01), while anxiety was associated with underweight and obesity adjusted by other dietary habits and psychological factors. Therefore, it is important to identify adolescents with overweight or obesity and establish prevention strategies for weight control in this age group, promoting healthy eating, physical activity and education in mental health.

## 1. Introduction

In recent decades, Mexico has gone through a nutritional transition, a process in which diets changed from being traditional and healthy, based on fruits, vegetables and whole grains, to industrialized with a high content of foods of animal origin and processed foods, rich in energy and saturated fats. In Mexico, these changes in dietary patterns are accompanied by an increase in the prevalence of overweight and obesity in childhood and adolescence [1].

The World Health Organization (WHO) defines adolescence as the age period between 10 and 19 years, which marks the separation between childhood and adult life. This period is characterized by rapid physical growth as well as cognitive and psychosocial development, which directly influence interaction with the environment and decision making [2].

Adolescence is divided into the following stages: early (10–14 years); middle (15–17 years old); and late (>18 years). During adolescence, important psychological and physical changes occur for the establishment of an individual’s lifestyle, which is a critical point for the consolidation of habits acquired during childhood and development [3].

Early adolescence is characterized by rapid physical growth and sexual development, which occur earlier in girls than in boys. Independence from parents begins and friends play an important role, self-interest in body image arises and identity is formed. The stage between 12 and 15 years old is characterized by having an unstable mood and low impulse control [4].

Overweight and obesity constitute the most important health problem in Mexico that affects the population from an early age until adulthood and is the main risk factor for the development of non-communicable or chronic degenerative diseases, such as diabetes, high blood pressure, dyslipidemia, cardiovascular diseases, osteoarticular diseases, certain types of cancer (breast, prostate and colon cancer), sleep apnea and depression, among others diseases [5].

Obesity in adolescents is a serious health problem in the world and in Mexico. The World Obesity Federation estimated that in 2020, there were 175 million schoolchildren and adolescents who were overweight and obese. It has been estimated that by the year 2030, there will be 310 million and, by the year 2035, 383 million individuals in these age groups [2]. In Latin America, 30% of the population aged 5 to 19 years has this condition, with Mexico, Argentina and Chile topping the list [6]. In case of Mexico, in adolescents, the prevalence of overweight and obesity between 2006 and 2021 increased from 33.2% to 42.9% [6,7], which is mainly due to the high consumption of added sugars at the national level. According to the study by Lopez-Olmedo et al. [8] in an adolescent population, 71% of men and 64% of women did not meet the dietary recommendation and their average consumption of added sugars was 72 and 64 g/day, respectively. In the National Health and Nutrition Survey 2020–2022 [7], it was reported that 62.3% of adolescents exceeded the recommended limit of consumption of added sugars (10% of total energy). The food groups that contributed the most to this high consumption were sweetened beverages, followed by snacks, sweets, desserts and sweet cereals.

Another factor that increases overweight and obesity is the sedentary lifestyle or physical inactivity, which is mainly associated with the use of screen technological media, where adolescents spend more than 3 h a day, whether watching television, playing video games or on the computer [9]. In the 2018–2019 National Health and Nutrition Survey [10], physical inactivity (according to the WHO classification) was reported in 46.3% of adolescents between 15 and 19 years old, as well as in 84.6% of adolescents between 10 and 14 years old. Similarly, in the study performed by Navarro et al. [11] on 470 Mexican adolescents, 74.3% of the participants had a low level of physical activity and 54.6% unhealthy eating habits.

Eating behavior is a human activity related to food intake and depends on internal and external factors of the individual. Emotional and behavioral problems are frequent in adolescents, which, in turn, may have an influence on eating behavior [12]. The prevalence of depression in Mexican adolescents from 12 to 17 years varies between 10 and 21.6%, which can reach 36.6% in women [13]. In adolescents, symptoms of depression and anxiety can be manifested as sleep and appetite disorders, learning difficulties, hyperactivity and irritability, as well as feelings of sadness and abandonment. Distress symptoms, such as palpitations, hyperventilation and diarrhea can also appear, generally associated with somatization processes, wherein obsessions and compulsions are considered more severe reactions of the process [12]. Ochoa et al. [14], in their research, which included cross-sectional longitudinal studies, systematic reviews and meta-analyses, reported that in children and adolescents aged 2 to 17 years, the prevalence of mild to severe anxiety and depression symptoms was high (37.4% and 43.7%, respectively).

The increasing prevalence of overweight and obesity among adolescents has become a critical public health problem that requires extensive research and intervention. Therefore, this study focuses on high school students in Mexico City, recognizing the need to understand the epidemiological panorama of this population group to carry out specific interventions and public health policies. In young population groups, early detection of mental health problems such as depression and anxiety and their relationship with eating patterns is crucial in order to prevent weight gain and the development of behavioral disorders of all types.

The objective of this study was to investigate the epidemiological panorama of overweight and obesity related to eating habits, physical activity and the concurrent presence of depression and anxiety in adolescents from high schools in Mexico City.

## 2. Methods

### 2.1. Study Design and Participants

A cross-sectional study was performed in the 2022–2023 school year, using a self-report questionnaire on eating habits and physical activity [15], as well as the Hospital Anxiety and Depression Scale [16], to explore potential links between psychological well-being and participants’ lifestyle choices.

This study was composed of 2710 participants and was performed on Mexican adolescents of both sexes between eleven and twelve years old, enrolled in 33 full-time high schools in Mexico City that formed part of the country’s educational system.

Following rigorous standards, all participants met specific inclusion criteria. A letter of informed consent, signed by the parents, and a letter of assent, signed by the adolescents themselves, were necessary to guarantee voluntary and informed participation in the study. Students were excluded if they met any of the following criteria: clinically confirmed diagnoses of diabetes, hypertension, metabolic syndrome or other chronic diseases, such as asthma or some type of cancer. It is important to mention that the educational system in Mexico does not allow biochemical samples to be taken from students within its facilities so as to know which obese students have metabolic syndrome; therefore, parents of the students were asked for their informed consent when they were sent information regarding whether their child had a confirmed diagnosis of the aforementioned physiological conditions. Additionally, adolescents who were pregnant or breastfeeding during the survey period, whose parents withdrew consent and who withdrew assent or refused to participate in any study-related activity were excluded from the study.

In this study, 33 high schools in Mexico City were selected, in which 2710 of them were first grade high school students enrolled in public high schools, with a median age of 12.02 years. High schools in the following municipalities were included: Álvaro Obregón, Azcapotzalco, Coyoacán, Cuauhtémoc, Gustavo A. Madero, Iztacalco, Iztapalapa, Miguel Hidalgo, Tláhuac, Tlalpan, Venustiano Carranza and Xochimilco.

#### Sample Size

To establish the size of the sample, a database of 119 schools in 16 municipalities from Mexico City was used, of which 46 technical high schools met the characteristics required for the study in the 2022–2023 school year (data provided by the Federal Education Authority of Mexico City).

The selection was based on simple random sampling with finite population and the formula by Murray and Larry [17] was used for the calculation:n = Zα^2^ · N · p · q
i^2^ (N−1) + Zα^2^ · p · q
where

n: Sample size.

N: Population size; a value of 46 is used (full-time high schools).

Z: Value corresponding to the approximately normal distribution, Zα = 1.62; α = 0.10.

p: Expected prevalence of the parameter to be evaluated, if unknown (p = 0.5), which increases the sample size.

q: 1 − p (if p = 50%, then q = 50%).

i: Error assumed (10%).

n = ((1.62)^2^ (46)(0.5)(0.5)/(0.01(46 − 1) + (1.62)(0.5)(0.5)) = 30.1806/0.85555 = 35.2 ≈ 33.

### 2.2. Ethics Committee

The study was approved by the Research Ethics Committee of the Division of Biological and Health Sciences of the Metropolitan Autonomous University with Agreement 7/21.5.4.

### 2.3. Socioeconomic Status Assessment

The Socioeconomic Levels Index, created by the Mexican Association of Market Intelligence and Opinion Agencies (AMAI) [18], classifies households using the “NSE Rule 2022”. This rule is an algorithm developed by the Socioeconomic Levels committee that measures the level of satisfaction of the most important needs of the home. This rule produces an index that classifies households into seven levels, considering the following six household characteristics:Education of the head of the household;Number of bedrooms;Number of complete bathrooms;Number of employed people aged 14 and over;Number of cars;Internet possession.

Each variable counts the scores, and at the end, they are added to obtain the following diagnosis:Up to 0 to 47 points = E (Very Extreme Low Level);Between 48 and 94 points = D (Extreme Low Level);Between 95 and 115 points = D + (Typical Low Level);Between 116 and 140 points = C − (Emerging Medium Level);Between 141 and 167 points = C (Typical Medium Level);Between 168 and 201 points = C + (Medium High Level);More than 202 points and more = A/B (High Level).

### 2.4. Anthropometric Measures

Anthropometric measurements of all adolescents from 33 schools were carried out daily at the same time, precisely from 8 to 9 a.m. To perform these measurements, all subjects wore light clothing, were barefoot, and met specific physiological conditions, including bowel movement, empty bladder, and a fasting period of at least 8 h.

Weight and height were measured using the reference anthropometric standardization manual [19] and obtained through the Inbody-270 body composition analyzer (South Korea). To assess weight status, the study used body mass index (BMI) percentiles recommended by the World Health Organization (WHO) [20].

The classification of adolescents who presented normal weight, overweight or obesity was carried out according to the BMI proposed by the World Health Organization (WHO); for the classification of BMI, age and sex were taken into account. Adolescents between the 5th and 85th percentile are defined as having normal weight; adolescents between the 85th and 95th percentile are defined as overweight; and adolescents with percentile ≥95 are defined as obese [21]. The calculations were executed using the Anthro Plus software 2009 (Geneva, Switzerland).

### 2.5. Eating Habits and Physical Activity Questionnaire

Self-report questionnaire on eating habits and physical activity, previously validated by Martínez et al. in Mexican adolescents, was applied [15,22], the reliability of this questionnaire was α = 0.778.

The time to answer the questionnaire lasted approximately 25 min with responses measured on a Likert scale (categories from 0 to 3) and consisted of four different sections, each of which focused on key aspects of the participants’ lifestyle:1.Eating habits (six questions):

The initial section evaluated eating habits, exploring factors such as frequency and quantity of food consumption. Six questions were used to obtain information about participants’ dietary patterns, with a total score of 12 points.

Some of the answer categories represented the frequency of food consumption per week: 0–2 days, 3–4 days, 5–6 days, and daily.

2.Intake of non-recommended foods (nine questions):

The second segment, composed of nine questions, evaluated the frequency and quantity of foods not recommended for daily intake. This section aimed to identify potential areas of nutritional concern (21 points).

Answer categories were 1 serving per day, 2 servings per day, 3 servings per day and 4 or more servings per day; and, regarding the frequency of consumption, the options were 0–1 day/week, 2–3 days/week, 4–5 days/week and 6–7 days/week.

3.Food environment and company (twelve questions):

The third section, which consisted of twelve questions, focused on the places where participants usually consumed their meals and the company they kept during mealtime. This exploration provided context to social and environmental influences on eating behaviors (18 points).

4.Physical activity (four questions):

The final section consisted of four questions and focused on physical activity.

Participants were asked about the frequency of their physical activity, both per day and per week. This section was dedicated to capturing information about their exercise routines (12 points).

Some answers were in relation to the duration of physical activity per day (less than 2 h/day, from 2 to less than 4 h/day, from 4 to less than 6 h/day and 6 or more hours/day) and about the frequency that they performed physical activities in days per week (0 to 2 times/week, 3 to 4 times/week, 5 to 6 times/week, and daily).

For the interpretation of eating and physical activity habits, these habits are classified according to the score achieved by the adolescent, compared with the maximum possible score, as shown (Table 1).

### 2.6. Hospital Anxiety and Depression Scale

With the purpose of evaluating symptoms of anxiety and depression, the Hospital Anxiety and Depression Scale (HADS) was used, previously validated in a Mexican population between 12 and 68 years old; Cronbach’s alpha of the total scale was 0.88 and of its two subscales >0.80 [16]. The scale is self-reported and allows us to identify symptoms of anxiety and depression. It consists of fourteen multiple-choice items and two subscales, depression and anxiety, each with seven items. The time to answer the questionnaire was approximately 15 min. The score for each subscale can vary between 0 and 21; each item presents four response options, ranging from absence or minimal presence = 0 to maximum presence of symptoms = 3. The higher the score obtained from each subscale, the greater the intensity or severity of symptoms: 0–7 points for absence of symptoms; 8–10 points for presence of symptoms; 11–21 points for complete clinical picture.

### 2.7. Statistical Analysis

Analyses were performed using STATA version 14 (Stata Corp., College Station, TX, USA). The Shapiro–Wilk normality test was used to evaluate the distribution of data in quantitative variables. Continuous quantitative variables with normal distribution were presented as mean and standard deviation, while those with non-parametric distribution were presented as median and 25th–75th percentiles as a measure of position. Frequencies and percentages were used to present categorical variables. A chi-squared test or Fisher’s F test was used to compare the study groups for categorical variables, and Student t or Mann–Whitney U test for unrelated samples or one-way ANOVA or Kruskal–Wallis test were performed for quantitative variables with three or more groups. Finally, the odds ratio (OR) was estimated using categorical variables through crosstabulations.

## 3. Results

During the study, there were no withdrawals of informed consent by parents or students. Consequently, the sample size remained at 2710 adolescents with a median age of 12.02 years [p25 11.1–p75 12.06]. Within this sample, 57.77% were female adolescents and 47.23% were male, with both sexes having an average age of 12.02 years [p25 11.1–p75 12.06]. No statistically significant differences were found (*p* = 0.680) in terms of height in general, observing an average of 1.55 m ± 0.07. When stratified by sex, the mean values were 1.53 m ± 0.05 for female adolescents and 1.57 m ± 0.08 for men, respectively (*p* < 0.001). Regarding weight, the general average was 50.7 kg [p25 43.2–p75 59.6]. Stratified by sex, the mean values were 49.75 kg [p25 43.0–p75 58.0] for female adolescents and 51.6 kg [p25 43.5–p75 62.2] for males (*p* < 0.001) (Table 2).

When evaluating anthropometric characteristics, no statistically significant differences were observed between BMI classifications (underweight, overweight, obesity and normal weight) in female adolescents (*p* = 0.450) or males (*p* = 0.794). However, both weight and height showed statistically significant differences for each sex (*p* < 0.05) using one-way ANOVA (Table 3).

In the evaluation of the prevalence of inadequate BMI by socioeconomic status (high, medium–high, medium, emerging medium, low, very low and extremely low), a trend was observed for each category. The highest frequency was consistently observed in individuals of medium socioeconomic status, followed by those in the medium–emerging and medium–high categories. When stratifying the data by sex, no associations were observed between male and female adolescents from different socioeconomic statuses (*p* = 0.224 and *p* = 0.878, respectively) (Figure 1).

Regarding eating habits, specifically the frequency of food consumption per week (categorized into 0–2 days, 3–4 days, 5–6 days and daily), the total number of female adolescents included in the study was 1430, while male adolescents represented 1280. No significant differences were observed in the frequencies of food consumption per week when comparing BMI diagnosis and sex, vegetables (*p* = 0.329 vs. 0.713), fruits (*p* = 0.427 vs. 0.380) and milk (*p* = 0.633 and 0.160) for female adolescents and male adolescents, respectively.

On the other hand, when evaluating the consumption of non-recommended foods, most of the items did not present associations with adolescents classified as overweight and obese by sex (females vs. males), sausages (*p* = 0.786 vs. 0.714), fast food (*p* = 0.263 vs. 0.459), bread and cookies (*p* = 0.298 vs. 0.854), snacks (*p* = 0.525 vs. 0.925), alcohol (*p* = 0.776 vs. 0.823) and sugary drinks (*p* = 0.093 vs. 0.327); however, it was found when evaluating the consumption of chocolates and sweets. This particular category showed a significant association with male adolescents classified as underweight, specifically those with a consumption frequency of 1 to 2 days. A difference was observed in the frequency of consumption according to diagnosis (*p* = 0.111 vs. 0.040).

Similarly, the analysis of vegetable portions consumed revealed that female adolescents did not present significant differences among the BMI diagnosis (*p* = 0.492). On the other hand, among male adolescents, the percentage distribution by BMI categories showed differences in the frequency of servings of vegetables per day. Specifically, those who had a lower consumption of vegetables (one serving per day) represented 25.81% in the underweight group, 37.70% consumed two servings per day in the overweight group, 36.10% consumed three servings per day in the group classified as obese and four or more daily servings were consumed by 7.66% in normal individuals, while individuals with normal weight had a higher frequency of consumption of vegetable servings compared with other groups, at 43.83% (*p* = 0.022).

Regarding fruit consumption, no significant differences were observed between male adolescents. However, in female adolescents, a similar trend was observed in the distribution of frequencies of fruit servings, with a statistically significant *p* value of 0.028 (Figure 2).

Regarding the frequency of days per week that individuals consume meals at different times (categorized as 0–1 day/week, 2–3 days/week, 4–5 days/week and 6–7 days/week), an evaluation was performed. Based on BMI and stratified by sex for breakfast time, no differences were observed between male and female adolescents. However, when stratifying by BMI, it was observed that male adolescents with obesity were more likely to skip breakfast compared with female adolescents, with a frequency of 0 to 1 day per week at 42.36% vs. 57.64% (*p* = 0.029) (Figure 3).

Regarding lunch and dinner, no differences were observed when stratifying by sex. However, among female adolescents diagnosed with underweight, a trend was observed in the frequency of consumption for each category of days. Specifically, underweight female adolescents notice a tendency to eat dinner less frequently compared with other BMI groups, although the difference did not reach statistical significance (*p* = 0.055) (Figure 4).

In the analysis of the physical activity section, the hours dedicated to these activities per day were assessed, categorized as less than 2 h/day, from 2 to less than 4 h/day, from 4 to less than 6 h/day and 6 or more hours/day with no differences when stratifying by BMI category and sex (female adolescents *p* = 0.549 vs. male adolescents *p* = 0.091). Similarly, the frequency of days a week during which these activities were carried out was evaluated, categorizing them as 0 to 2 times/week, 3 to 4 times/week, 5 to 6 times/week and daily; no differences were found (female adolescents *p* = 0.616 vs. male adolescents *p* = 0.831).

Additionally, an evaluation was performed to explore the possible relationship between mealtime and usual place of food consumption with the diagnosis of BMI. Consistent trends were identified for breakfast and lunch across all groups, with home being the predominant location for food consumption, followed by outside with food brought from home. No statistically significant differences were detected between the groups for both female adolescents and male adolescents (breakfast: 0.934 vs. 0.913, lunch: 0.236 vs. 0.837, respectively).

Regarding dinner time, normal-weight female adolescents were found to predominantly eat dinner at home compared with individuals in other BMI categories, and this difference reached statistical significance (*p* = 0.05).

Prevalence was calculated within the study sample, revealing an overall underweight rate of 6.7%. Stratifying by sex, underweight occurred in 6.3% of female adolescents and 7.2% of male adolescents. Regarding overweight, the prevalence was 26.5%, with rates of 28.3% in female adolescents and 24.4% in male adolescents. Obesity had a global prevalence of 20.0%, with 16.0% in female adolescents and 24.4% in male adolescents.

Additionally, odds ratios (ORs) were estimated to evaluate the association of these factors with the prevalence of underweight, overweight and obesity compared with those of normal weight. The analysis was adjusted by sex, considering diagnoses related to eating habits and physical activity (adequate and inadequate).

The findings indicated that female adolescents with inadequate eating habits were associated with having obesity (OR = 1.95; CI 1.009–3.76) (Table 4).

Evaluating the association between consuming meals alone at specific times (such as breakfast, lunch or dinner) and a positive diagnosis of depression and anxiety, female adolescents with obesity exhibited an association with increased depression at any meal time, while anxiety was related to female adolescents who were both overweight and obese. In male adolescents who were overweight and obese who ate breakfast alone, an association was found with both depression and anxiety, in contrast to those who ate with others (Table 5).

Additionally, associations were observed between depression symptoms and obesity (OR = 5.68, CI 1.36–32.81; *p* = 0.01), while anxiety was associated with underweight and obesity adjusted by other dietary habits and psychological factors.

## 4. Discussion

In the present study, it was found that the prevalence of overweight and obesity in adolescents was 46.5% (26.5% overweight and 20.0% obese), higher in female adolescents (28.3%) than in male adolescents (24.4%). However, in the case of obesity, male adolescents had a higher prevalence (24.4%) compared with female adolescents (16.0%).

Likewise, in the 2021 National Health and Nutrition Survey [23], it was found that the prevalence of overweight and obesity in the population aged 12 to 19 was 42.9% (24.7% overweight and 18.2% obese), with the percentage being higher for overweight female adolescents (26.4%) compared with male adolescents (23.0%). However, in the case of obesity, male adolescents had a higher prevalence (21.5%) compared with female adolescents (15.0%) and this excess weight in adolescence is closely linked to eating habits and physical activity.

In Mexico, there has been an alarming growth of overweight and obesity in adolescents, Shamah et al. [6] mentioned that in 1988, the prevalence of overweight and obesity in female adolescents was 11.1%, and in 2018, it increased to 38.4%. In the case of male adolescents in this same age group, the prevalence of overweight and obesity in 2006 was 33%, and in 2018, it was 35.6%. Additionally, in the National Health and Nutrition Survey 2020–2022, it has been reported that the prevalence of overweight and obesity in adolescents reached 41%, which means a 50% increase between 2006 and 2020–2022. Similarly, according to worldwide statistics, in the last 30 years, the increase in the prevalence of overweight and obesity in children and adolescents has been alarming [24].

In Mexico, the prevalence of this condition in adolescents residing in rural areas was lower; however, in recent years, in student populations, it has increased more markedly, especially among the disadvantaged and minority groups that suffer from high levels of social vulnerability, particularly in the center and south of the country [25].

Among the main causes of obesity in Mexican adolescents are an excess consumption of foods and drinks with high energy density (fat and sugars has been documented as a substitute for natural foods), as well as prolonged time in front of screens and a lack of physical activity. Furthermore, adolescents are immersed in an obesogenic environment that predominates at school and home [26].

Pérez-Herrera et al. [21] mentioned that the panorama of overweight, obesity and diabetes in Mexico is explained in part by the nutritional transition that the country is experiencing, i.e., a westernization of the diet, characterized by (a) an availability of low-cost processed foods, added with high amounts of fat, sugars and salt; (b) an increase in fast food and food prepared outside the home; (c) a reduction in time available for food preparation; (d) an exposure to advertising about industrialized foods and products that facilitate people’s daily tasks and work, which reduce their energy expenditure; and (e) decreased physical activity.

In the present study, a low consumption of vegetables and fruits was found (which are considered foods in the recommended groups for daily consumption) and that adolescents of normal weight had a higher frequency of consumption of vegetables compared with other groups. This is possibly related to the fact that in individuals with normal weight, the fiber content presented in vegetables and fruits gives a greater feeling of satiety and low caloric density, which, at the same time can reduce the intake of foods rich in fats and sugars. On the other hand, a high consumption of sausages, fast food, breads, cookies, snacks and sugary drinks in the groups was found. However, there was no association between the consumption of sugary drinks and overweight or obesity—similar to the data reported by Shamah et al. [26]—which implies that more research is required to understand why no relationship has been found between overweight, obesity and intake of free sugars in adolescents.

Similar to our study, Gaona et al. [27] indicated that according to the National Health and Nutrition Survey 2020–2022, a low daily consumption of healthy foods was observed: only 31.4% of adolescents consumed vegetables, 39.1% fruits, and 46.8% dairy products. With respect to the groups of foods not recommended for daily consumption, 89.9% of adolescents consumed sweetened drinks, 43.4% snacks, sweets and desserts, as well as 34.2% consumed fast food and fried Mexican snacks. Likewise, in the study by Lopez-Olmedo et al. [8] conducted on Mexican adolescents, the majority of participants exceeded dietary recommendations for daily sugar consumption, and their intake of sweetened foods was high.

Additionally, Medina-Zacarías et al. [28], in their study on 732 Mexican adolescents, showed that participants who had an unhealthy eating pattern presented a greater probability of being overweight or obese, compared with those who have healthier diets. The results of the study conducted by Turnbull et al. [29] informed that Mexican teenagers and parents prefer an unhealthy diet since they have easy access to cheap processed foods and junk foods, which may be related to family activities or represent family unity. This indicates how globalization in developing countries has allowed industrialized food marketers to introduce a wide variety of products, distribute them widely, price them reasonably, adapt them to local cultures and promote them to consumers.

Both in Mexico and internationally, the consumption of foods and beverages with high energy density, fat content and added sugars have been replacing natural foods. Likewise, it has been argued that the commercialization of these unhealthy products affects preferences, leading children and adolescents to excessive weight [30]. Food preferences in this population tend to be sweet and salty flavors, as opposed to acidic and bitter, a fact that favors the intake of foods rich in fat and sugar, and subsequently rejecting healthier foods, such as fruits and vegetables [31].

Brambila et al. [32] stated that the mechanism through which Mexican families may affect the weight of their children is an acceptance of a low level of physical activity and unhealthy eating patterns, including family meal times that have an important influence on overweight and obesity. Guerrero et al. [33] reported that to reduce factors associated with obesity, it is essential to intervene in routine acquisition, which includes establishing eating schedules, exercising regularly, reducing sedentary lifestyles and establishing sleep routines.

In the present study, it was observed that male adolescents with obesity were more likely to skip breakfast compared with female adolescents, with a frequency of 0 to 1 day a week (42.36% vs. 57.64%); and regarding lunch and dinner, malnourished female adolescents showed a tendency to eat dinner less frequently compared with the other BMI groups. Likewise, regarding dinner time, normal-weight female adolescents were found to predominantly eat dinner at home compared with adolescents in other BMI categories.

The findings of the present study indicated that being a woman with inadequate eating habits was associated with obesity (OR = 1.95). Likewise, in a study performed by Avalos et al. [34], female adolescents with high waist/hip values stated that they ate food even when they were satiated and of which they were aware. In the study by Aceves et al. [35], some Mexican parents reported that they do not expect leftovers on their children’s plates, even if they are not hungry, they have to finish the entire portion that was served to them. This unhealthy eating behavior may affect their weight and be a risk factor for future psychological alterations such as anxiety, depression or body dissatisfaction.

On the other hand, in the analysis of physical activity in the present study, the hours dedicated to this activity per day were assessed, and no differences when stratifying by BMI category and sex were found. Likewise, the frequencies of days per week during which these activities were performed were evaluated and no differences were found between female and male adolescents.

Regarding a sedentary lifestyle, Soltero et al. [36] recommend that children and adolescents should limit themselves to less than or equal to 2 h of screen time per day. However, the 2018 National Health and Nutrition Survey [26] reported physical inactivity in 46.3% of adolescents between 15 and 19 years old, as well as in 84.6% of adolescents between 10 and 14 years old. Additionally, 77.3% of children and Mexican adolescents exceeded the 2 h screen time limit, and since screen time significantly contributes to obesity, there is an urgent need to increase our understanding of how screen time impacts dietary behaviors and physical activity related to obesity among Mexican adolescents.

Analyzing the lifestyles of young individuals is of great importance, considering that in Mexico, the prevalence of overweight and obesity is increasing in the population aged 12 to 15 years with a high risk of complications in their adult life, generating a loss of years of healthy life and an increased cost in health services [37].

The high prevalence of depression (34.2%, 37.8% in female and 30.6% in male adolescents) and anxiety (47.1%, 49.6% in female and 44.5% in male adolescents) symptoms was found in the present study, more in girls that in boys.

These data coincide with the results published both nationally [13] and internationally [14] in this age group. According to Gomez-Peresmitre [38], 78% of Mexican adolescents from high schools were at risk of depression. Gonzalez et al. [39] showed that school environment, relationship with friends, neighborhood function, poor living conditions and family structure, as well as poor emotional self-regulation are risk factors of this situation. Alvarez et al. [40] found that family climate and available external resources to support adolescents were protective factors to prevent depressive disorder, which are crucial for their emotional well-being, academic performance and interpersonal relationships. Additionally, it has previously been reported that eating patterns are related to emotion regulation, and maladaptive anxiety managing may lead to eating disorders and weight gain [41,42].

In the present study, it was found that adolescents with obesity had an association with depression, which may be due to diverse causes as they eat their meals outside of their homes at a specific moment or at every mealtime, or eat alone no matter the place, while adolescents who presented anxiety had an association with both underweight and obesity.

Likewise, Meza Peña et al. [43], in their study, showed that adolescents who are overweight or obese have greater depressive symptoms than those with normal BMI. In the same way, Moreno et al. [44] found that about a third of adolescents presented some level of depression, predominantly in the female group. A mild level of mood disturbance was more prevalent in female adolescents; regarding moderate depression, it occurred in equal proportions in both sexes. The only case of extreme depression was in the female sex. Gutierrez et al. [45] mentioned that overweight adolescents are more likely to have a negative body image and low self-esteem, which makes them lean more toward food, possibly as a source of comfort in moments of mood disturbance. Excess body fat in adolescents may cause a variety of clinical and psychosocial conditions since those with obesity could suffer discrimination, low self-esteem and depression [46]. In the study by Sánchez et al. [47], an association between self-image, self-esteem, depression and the presence of obesity in Mexican schoolchildren and adolescents has been shown.

The World Health Organization (WHO) in 2021 [48] estimated that one in seven adolescents aged 10 to 19 years (14%) worldwide suffers from a mental disorder, and that depression, anxiety and behavioral disorders are among the leading causes of illness and disability between adolescents, and yet, these illnesses remain largely unrecognized and untreated. According to the National Health and Nutrition Survey 2018–2019 [10], despite the high prevalence of depression in adolescents in Mexico, only 3.7% of participants had a medical diagnosis of this condition.

In young population groups, early detection of mental health problems is crucial for treatment and prognosis, and the exploration of the implications of these findings on the eating habits remains ongoing, as well as their role on the development of behavioral disorders of all types.

Various strategies should be considered to help prevent symptoms of depression and anxiety in adolescents, such as promoting open communication with their guardians and parents, teaching problem-solving and positive thinking skills, as well as effective management of anxiety and challenges. Lifestyle changes that involve healthy eating, physical exercise and quality sleep will allow adolescents to develop strategies that help them cope with adversity in a more adaptive way.

## 5. Limitations

Among the limitations of this study was its cross-sectional design, which does not allow for the causal interpretation of the associations identified by the statistical models. Additionally, self-reported questionnaires were administered, which are susceptible to response bias. The questionnaires applied in the sample to evaluate anxiety and depression identified symptoms, but did not diagnose any clinical condition.

However, one of the strengths of this study was the large sample size, which included adolescents from 33 high schools in Mexico. Additionally, precise anthropometric measurements were performed and all instruments applied in the present study were previously validated in different population groups, including Mexican adolescents, and are frequently used to identify unhealthy behaviors at the national level. Further research is required to assess underlying factors of the obesogenic environment, which may help develop targeted and coordinated prevention programs in young population groups. Finally, longitudinal studies are needed to evaluate the long-term effect of depression and anxiety on body weight.

## 6. Conclusions

A high prevalence of overweight (26.5%) and obesity (20%) was found in Mexican high schools in Mexico City. It is necessary that studies do not continue by only considering the known risk factors such as the quantity and type of food, which is undoubtedly important since the nutritional quality between a natural food and an ultra-processed one is different. However, it is also necessary to take into account aspects related to these habits such as who they eat their meals with, if their meals are homemade or if they eat at home, especially at this age, where the sense of social belonging is important because that will determine much of their psychological and emotional development, which can influence eating behaviors such as overeating and contributing to the prevalence of overweight and obesity.

This is how it has been pointed out in the results of this work, where physical activity is considered a typically very important factor that contributes to the problem of overweight and obesity among adolescents, but this was only observed in girls who had almost double the risk of being obese with inadequate physical activity habits (OR = 1.95). On the other hand, when factors such as eating their food alone are considered, it was observed that among girls, the risk of having depression and obesity can range from 3.31 to 7.00 times that of having obesity. While for boys, it was between 2.12 and 5.90 times the risk of having obesity.

Therefore, it is important to develop strategies aimed not only at promoting healthy eating and physical activity in all environments in which adolescents interact, but to also pay attention to the mental health of this population group, especially to the behaviors and attitudes presented in the educational field and also at home. Collaboration between the health and education sectors, the government and the family is essential. It is also important to address this problem effectively.

## Figures and Tables

**Figure 1 healthcare-12-00604-f001:**
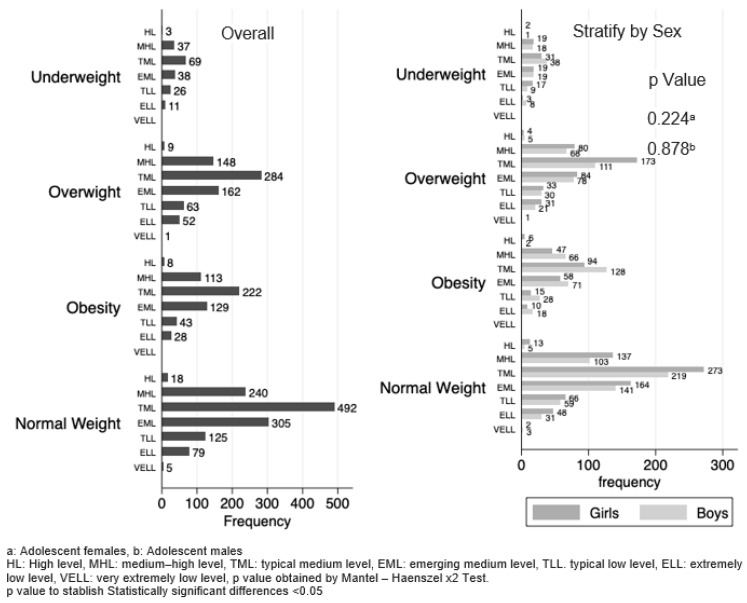
Socioeconomic status by BMI diagnosis stratified by sex.

**Figure 2 healthcare-12-00604-f002:**
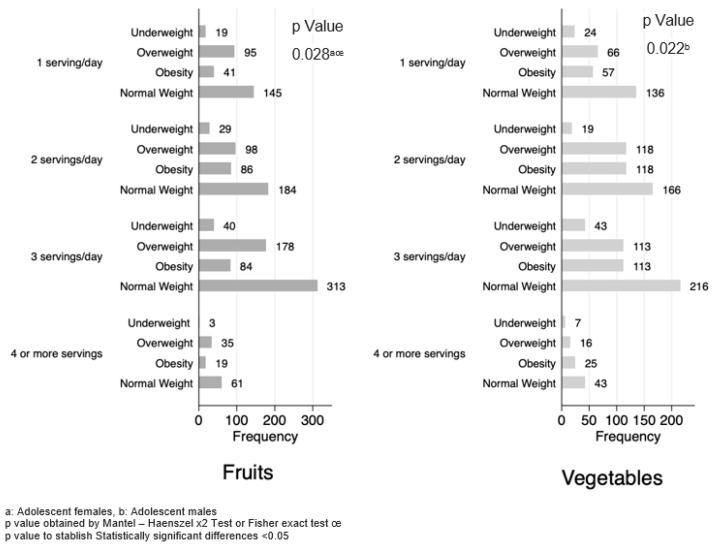
Frequency of food portions by BMI diagnosis stratified by sex.

**Figure 3 healthcare-12-00604-f003:**
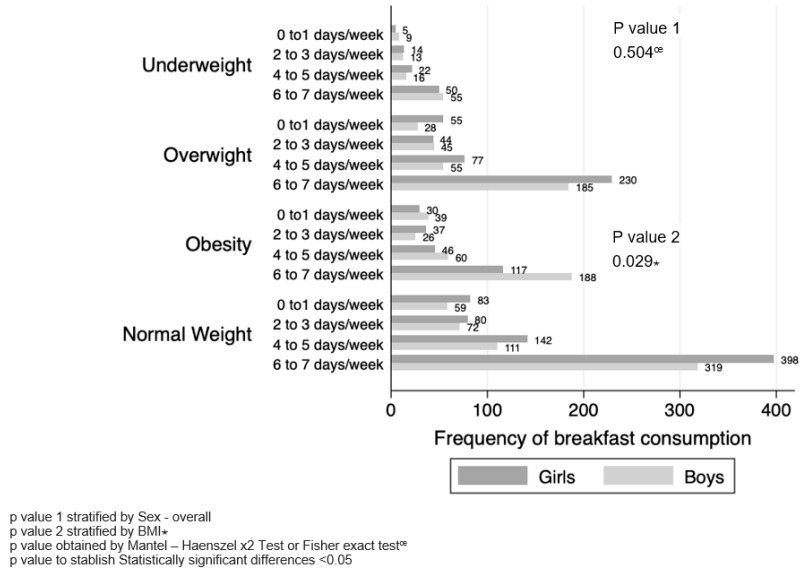
Frequency of breakfast consumption by BMI stratified by sex.

**Figure 4 healthcare-12-00604-f004:**
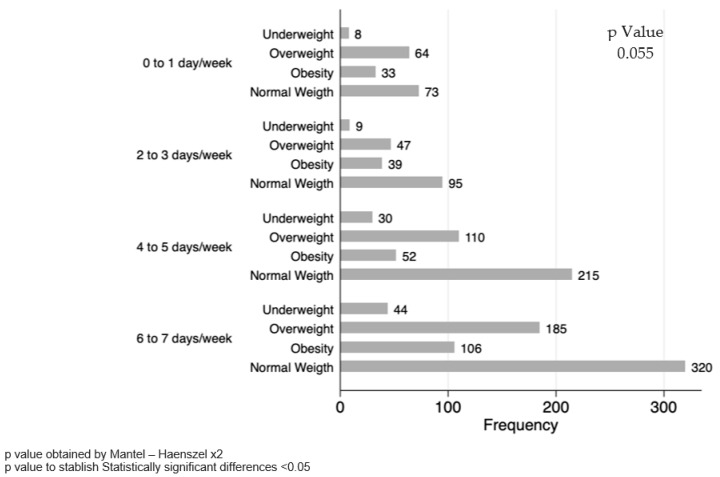
Frequency of days per week that adolescent females have dinner.

**Table 1 healthcare-12-00604-t001:** Criteria for the classification of eating and physical activity habits [15].

Variable	Category	Designation	Criteria	Score
Eating habits	1	Inadequate	<50% of the maximum possible score	<25.5
	2	Partially inadequate	≥50% and <75% of the maximum possible score	≥25.5 and <38.5
	3	Adequate	≥75% of the maximum possible score	≥38.5
Physical activity habits	1	Inadequate	<50% of the maximum possible score	<6
	2	Partially inadequate	≥50% and <75% of the maximum possible score	≥6 and <9
	3	Adequate	≥75% of the maximum possible score	≥9

**Table 2 healthcare-12-00604-t002:** Anthropometric characteristics, habits, and psychological diagnosis by sex.

Variable	n = 2710	Females n = 1430	Males n = 1280	*p* Value
Age (y)	12.02 [11.1–12.06]	12.02 [11.1–12.06]	12.02 [11.1–12.06]	0.680
Height (m)	1.55 ± 0.07	1.53 ± 0.05	1.57 ± 0.08	<0.0001 ^¥^
Weight (kg)	50.7 [43.2–59.6]	49.75 [43.0–58.0]	51.6 [43.5–62.2]	<0.001 ^‡^
BMI diagnosis				
Underweight	6.79%	6.36%	7.27%	<0.001 ^‡^
Overweight	26.53%	28.39%	24.45%	
Obesity	20.04%	16.08%	24.45%	
Normal weight	46.64%	49.16%	43.83%	
Eating habits				
Inadequate	10.00%	10.70%	9.22%	0.245
Partially inadequate	76.86%	76.92%	76.80%	
Adequate	13.14%	12.38%	13.98%	
Physical activity habits				
Inadequate	46.35%	48.11%	44.38%	0.057
Partially inadequate	35.46%	35.17%	35.78%	
Adequate	18.19%	16.71%	19.84%	
Depression diagnosis				
Depressed	10.04%	11.54%	8.36%	<0.001 ^‡^
Probably Depressed	24.39%	26.29%	22.27%	
Normal	65.75%	62.17%	69.38%	
Anxiety diagnosis				
Anxiety	24.21%	25.38%	22.89%	0.026
Probably Anxiety	22.99%	24.27%	21.56%	
Normal	52.80%	50.35%	55.55%	

Mean (±standard deviation), median [25th–75th percentile]. *p* value: Student *t* test for difference between groups or Mann–Whitney U test. *p* value to establish statistically significant differences <0.05, <0.001 ^‡^, <0.0001 ^¥^.

**Table 3 healthcare-12-00604-t003:** Anthropometric characteristics by sex according to the BMI diagnosis.

**Adolescent Females**					
**Variable**	**Underweight** **91**	**Overweight** **406**	**Obesity** **230**	**Normal Weight** **703**	***p*****Value** ^**2**^
Age (y) *p* value ^1^	12.03 [11.11–12.06] 0.409	12.01 [11.10–12.06] 0.227	12.02 [11.10–12.06] 0.775	12.02 [11.10–12.06]	0.450
Height (mts) *p* value ^1^	1.52 ± 6.98 1.000	1.54 ± 5.44 <0.01 ^†^	1.55 ± 6.0 <0.001 ^‡^	1.52 ± 5.8	<0.0001 ^¥^
Weight (kg) *p* value ^1^	36.5 [33.5–39.5] <0.0001 ^¥^	68.4 [63–74.8] <0.0001 ^¥^	55.5 [51.9–59.5] <0.0001 ^¥^	44.6 [41.3–48.4]	<0.001 ^‡^
**Adolescent Males**					
	**Underweight** **93**	**Overweight** **313**	**Obesity** **313**	**Normal Weight** **561**	***p* Value ^1^**
Age (y) *p* value ^2^	12.02 [12.00–12.05] 0.309	12.02 [11.10–12.06] 0.616	12.01 [11.10–12.06] 0.916	12.02 [11.10–12.06]	0.794
Height (mts) *p* value ^2^	1.53 ± 9.81 1.000	1.57 ± 8.49 0.001 ^†^	1.59 ± 7.62 0.0001 ^‡^	1.56 ± 8.53	<0.0001 ^¥^
Weight (kg) *p* value ^2^	36.5 [33.5–39.5] <0.01 ^†^	68.4 [63–74.8] 0.236	55.5 [51.9–59.5] <0.001 ^‡^	44.6 [41.3–48.4]	<0.001 ^‡^

Mean (±standard deviation), median [percentile 25th–75th]. *p* value ^1^: ANOVA one-way test for difference between groups. *p* value ^2^: Bonferroni post hoc difference with normal weight group. *p* value to establish statistically significant differences <0.05, <0.01 ^†^, <0.001 ^‡^, <0.0001 ^¥^.

**Table 4 healthcare-12-00604-t004:** Diagnosis of underweight, overweight or obesity associated with the diagnosis of inadequate eating habits or physical activity habits.

**Adolescent Females**				
**Underweight**				
**Variable**	**With**	**Without**	**OR**	**CI**
Inadequate ^a^	7	68	1.5	0.50–4.46
Adequate	7	102		
Inadequate ^b^	47	323	1.71	0.86–3.41
Adequate	11	130		
**Overweight**				
Inadequate ^a^	52	68	1.62	0.98–2.67
Adequate	48	102		
Inadequate ^b^	188	323	1.16	0.82–1.64
Adequate	65	130		
**Obesity**				
Inadequate ^a^	26	68	1.95 ^†^	1.009–3.76
Adequate	20	102		
Inadequate ^b^	130	323	1.24	0.79–1.93
Adequate	33	130		
**Adolescent Males**				
**Underweight**				
**Variable**	**With**	**Without**	**OR**	**CI**
Inadequate ^a^	7	53	0.80	0.30–2.14
Adequate	13	79		
Inadequate ^b^	37	247	0.82	0.44–1.51
Adequate	18	99		
**Overweight**				
Inadequate ^a^	24	53	0.70	0.38–1.27
Adequate	51	79		
Inadequate ^b^	140	247	0.79	0.54–1.14
Adequate	71	99		
**Obesity**				
Inadequate ^a^	34	53	1.40	0.78–2.52
Adequate	36	79		
Inadequate ^b^	66	247	0.80	0.49–1.29
Adequate	33	99		

^a^: Eating habits. ^b^: Physical activity habits. *p* value obtained by Mantel–Haenszel Chi^2^ test or Fisher’s exact test. Significance *p* < 0.05, <0.01 ^†^.

**Table 5 healthcare-12-00604-t005:** Diagnosis of BMI associated with eating alone in different moments and having depression or anxiety.

**Adolescent Females**				
**Breakfast**				
**Variable**	**With**	**Without**	**OR**	**CI**
W/Underweight ^a^	4	13	2.62	0.81–8.50
W/Overweight	24	99	2.06	1.16–3.65 ^‡^
W/Obesity	18	29	5.29	2.66–10.52 ^§^
Normal weight	34	290		
W/Underweight ^b^	10	9	3.10	1.21–7.91 ^‡^
W/Overweight	56	67	2.88	1.86–4.46 ^§^
W/Obesity	32	22	5.01	2.75–9.13 ^§^
Normal weight	76	262		
**Lunch**				
W/Underweight ^a^	1	8	1.15	0.14–9.47
W/Overweight	4	30	1.23	0–41–3.66
W/Obesity	10	13	7.11	7.11–17.20 ^§^
Normal weight	43	398		
W/Underweight ^b^	7	4	5.75	1.65–20.04 ^‡^
W/Overweight	13	26	1.64	0.81–3.31
W/Obesity	17	8	6.99	2.93–16.64 ^§^
Normal weight	107	352		
**Dinner**				
W/Underweight ^a^	3	12	1.98	0.54–7.31
W/Overweight	9	56	1.27	0.59–2.75
W/Obesity	10	24	3.31	1.48–7.37 ^‡^
Normal weight	45	358		
W/Underweight ^b^	9	5	5.57	1.82–16.99 ^‡œ^
W/Overweight	22	41	1.66	0.94–2.91
W/Obesity	18	17	3.26	1.61–6.64 ^¥^
Normal weight	105	325		
**Adolescent Males**				
**Breakfast**				
**Variable**	**With**	**Without**	**OR**	**CI**
W/Underweight ^a^	3	14	3.78	0.97–14.70
W/Overweight	19	108	3.10	1.50–6.41 ^¥^
W/Obesity	23	105	3.86	1.91–7.95 ^§^
Normal weight	14	247		
W/Underweight ^b^	6	12	2.12	0.76–5.93
W/Overweight	42	83	2.15	1.33–3.48 ^‡^
W/Obesity	24	48	2.12	1.19–3.78 ^‡^
Normal weight	51	217		
**Lunch**				
W/Underweight ^a^	0	2	-	-
W/Overweight	3	34	1.36	0.38–4.81
W/Obesity	9	33	4.22	1.78–9.96 ^¥^
Normal weight	21	325		
W/Underweight ^b^	4	2	7.45	1.03–41.46 ^†^
W/Overweight	8	22	1.35	0.58–3.16
W/Obesity	17	20	3.16	1.58–6.35 ^‡^
Normal weight	77	287		
**Dinner**				
W/Underweight ^a^	3	8	6.62	1.62–27.20 ^†^
W/Overweight	5	43	2.06	0.72–5.83
W/Obesity	13	39	5.90	2.68–12.97 ^§^
Normal weight	18	319		
W/Underweight ^b^	6	6	4.17	1.30–13.36 ^‡œ^
W/Overweight	12	32	1.56	0.76–3.20
W/Obesity	23	27	3.56	1.92–6.59 ^§^
Normal weight	67	280		

W: With. ^a^: Depression. ^b^: Anxiety. *p* value obtained by Mantel–Haenszel Chi^2^ test or Fisher’s exact test. *p* value to establish statistically significant differences, <0.01 ^†^, <0.001 ^‡^, <0.0001 ^¥^, <0.00001 ^§^. ^œ^ is for those who had the Fisher Exact test performed.

## Data Availability

Data are contained within the article.

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
