# Peer review of "Epidemiological Overview of Overweight and Obesity Related to Eating Habits, Physical Activity and the Concurrent Presence of Depression and Anxiety in Adolescents from High Schools in Mexico City: A Cross-Sectional Study"

_healthcare, 2024, doi:10.3390/healthcare12060604_

Round 1

Reviewer 1 Report

Comments and Suggestions for Authors

Section: Introduction

The objective is well defined, and the introduction adequately introduces the problem and background of the research. However, there is a need to include information regarding depression in the specified age group.

Other specific issues that need attention:

·       Page 2, Lines 48-74: The cited references are in Spanish, hindering the assessment of accuracy. Please provide translations or summaries for evaluation.

Section: Methods

This section presents significant methodological issues, primarily concerning the validity of the questionnaires. It is advisable to conduct a separate validation study for a different age group, assessing reliability and validity within the context of the group's characteristics and developmental stage. Cronbach's alpha, while indicating internal consistency within a sample, does not ensure reliability across age groups. Further validation work is required to establish the instrument's appropriateness and reliability in diverse populations. Please address this concern appropriately.

Other specific issues that need attention:

·       Page 2, Lines 91-93: Clearly define the criteria for school and student participation.

·       Page 3, Lines 97-99: The exclusion criteria state that students with confirmed metabolic syndrome were excluded from the current study. Metabolic syndrome is defined as the presence of obesity, indicated by either a waist circumference >= 88 cm in women and >= 102 cm in men or a body mass index >=30 kg/m2, along with two of the following three criteria: high normal blood pressure or hypertension, impaired glucose metabolism, and elevated non-high-density lipoprotein (non-HDL) cholesterol level (atherogenic dyslipidaemia), according to the provided references (https://doi.org/10.1007/978-3-319-48382-5_1, https://doi.org/10.5114%2Faoms%2F152921). Given that metabolic syndrome encompasses obesity, it raises a question about the exclusion of obese children. A clear rationale for this apparent discrepancy is needed. Additionally, please provide clarification on how the defined criteria for metabolic syndrome apply to children.

·       Page 3, Lines 98-99: Provide a full list of chronic diseases excluded.

·       Page 3, Lines 131: Explain how the assessment of satisfactory bowel movement was conducted.

Section: Discussion

·       Page 12, Lines 518-520: Elaborate further on the provided statement, explaining the association in more detail.

Section: Limitations and Study Strengths

These sections are entirely missing. Please include a discussion on the limitations and strengths of the study.

Author Response

We appreciate the observations made by the Reviewer 1 that allowed us to substantially strengthen the document and we send the responses to each of them.

Reviewer 2 Report

Comments and Suggestions for Authors

The article "Epidemiological overview of overweight and obesity related to eating habits, ..." examines the prevalence of overweight and obesity, physical inactivity, dietary habits in Mexican adolescents; and also associations of these previous factors with depression.

In my view, the article is of very poor scientific quality and does not merit be published in Healthcare or been re-submitted.

There are many weak points that explain this negative evaluation.

First, authors do not provide enough details about how the sample was obtained. This information is critical to understand the validity of the study.

Second, I am surprised that authors do not employ valid methods of measurement of dietary habits or physical inactivity; and none reference is inserted in methods.  This limitation cannot be changed with a new wave of reviews as the study was finished.

Third, the study is a cross-sectional study. Performing associations with mental health variables does not help to increase the scientific evidence about how mental health is shaped by environmental factors in adolescente. 

Fourth, the article is poorly written. For example, the introduction is full with mentions of prevalence data but the reader does not know whether prevalences are describing: overweight, obesity, or both.

Fifth, the introduction is not well justified and contextualised. There are not mentions to similar previous studies conducted in Mexico in adolescents.

Sixth, it is unclear why authors are using percentiles but they do not describe the cut-points in the text.

Seven, some figures include information in Spanish language. Ej. sex of participants.

In general, there are an excessive number of methodological, intellectual and writing problems in the submitted article. I regret to tell the authors that this article has not enough scientific quality to be published in Healthcare journal (JCR with Q2 impact factor). 

Comments on the Quality of English Language

English language can improve but it is not a major problem in this article.

Author Response

We appreciate the observations made by the reviewer 2 that allowed us to substantially strengthen the document and we send the responses to each of them.

Reviewer 3 Report

Comments and Suggestions for Authors

Introduction

It is necessary to add a greater number of citations to support the information provided.

They talk about obesity, the numbers in Mexico, the problems they bring, factors that increase it, among others. In the penultimate paragraph they talk about secondary students, but nothing more is said about them. Information about its characteristics should be added here, which is what the scientific evidence so far says about the study variable in this population.

Then the research problem should be submitted. Arguments that indicate the need to carry out the research.

When reading the last paragraph, I see that other variables are also discussed, which is why information about them must also be provided, and in particular what the scientific evidence says in the group of people to be studied.

Methods

I don't understand the issue of depression perfectly, but I see that the instrument is to see anxiety and depression, is it the same? If they are the same, or not, or if they are related, they should explain it in the introduction.

Results and discussion are well described.

When reading the conclusion I have some doubts, since the objective is described, and this is not the same as the one that appears at the end of the introduction.

As a final review, I think they should add more information in the introduction, small adjustments in the methodology and discussion, and the conclusion should be rewritten taking into account what the objective is.

Author Response

We appreciate the observations made by the reviewer 3 that allowed us to substantially strengthen the document and we send the responses to each of them.

Reviewer 4 Report

Comments and Suggestions for Authors

I recommend using the STROBE checklist for a more robust report.

Realizing the degree of importance in a cross-sectional study, I indicate the need to report two methodological issues: (I) whether the sample is a census involving the total number of students, or by some probabilistic technique or by convenience; (II) whether the sample is representative - and, if so, which public does it represent (adolescents enrolled in public high school)?

Another point I consider important in a cross-sectional study: was the person conducting the analysis blinded to the exposures and outcomes?

I suggest using grayscale in the bars of the Figures, thinking of grayscale prints.

The discussion needs to be more objective and in-depth, based on the study's main evidence.

It is important to point out the limitations and potential of the research, as well as avenues for future studies.

What is the Gonzalo Río Arronte Foundation's involvement in the research? It is necessary to indicate the foundation's objective; its degree of participation throughout the research, as well as in the presentation of the results.

Author Response

We appreciate the observations made by the reviewer 4 that allowed us to substantially strengthen the document and we send the responses to each of them.

Round 2

Reviewer 1 Report

Comments and Suggestions for Authors

Thank you for your recent revisions and the effort you have put into addressing the comments from the previous review. However, after careful consideration, it has become apparent that further revisions are necessary before your research can be accepted for publication.

Firstly, I must point out a significant conceptual discrepancy in your initial response. You appear to have conflated the terms 'anxiety' and 'stress,' which, in psychological terms, have distinct definitions and implications. This confusion needs to be clarified to maintain the academic rigor of your study.

Additionally, I have concerns regarding the validation of your questionnaire, as mentioned in your third response. The study you have cited for validation purposes is not available in English, which unfortunately limits my ability to thoroughly assess the validity of your claims regarding the questionnaire's efficacy. This is a critical aspect of your research that requires further elucidation.

Another point of concern is your use of participants who are under 12 years of age. You have noted that the questionnaire has been validated for this age group, but given the aforementioned issue with the validation study's accessibility, this claim remains unsubstantiated. It is imperative that this issue be addressed to ensure the reliability and ethical compliance of your research.

In light of these issues, I recommend another round of revision. It is essential that these concerns are adequately addressed to meet the publication's standards. I appreciate your dedication to this research and look forward to seeing the revised manuscript.

Author Response

Response to Reviewer 1.

Comments and Suggestions for Authors

Thank you for your recent revisions and the effort you have put into addressing the comments from the previous review. However, after careful consideration, it has become apparent that further revisions are necessary before your research can be accepted for publication.

We thank the reviewer for all the observations made to the manuscript that allowed it to be improved.

Point 1:

Firstly, I must point out a significant conceptual discrepancy in your initial response. You appear to have conflated the terms 'anxiety' and 'stress,' which, in psychological terms, have distinct definitions and implications. This confusion needs to be clarified to maintain the academic rigor of your study.

Response 1

In the introduction section, Ochoa et al analyzed studies that evaluated stress, behavioral disorders, anxiety, depression, suicidal risk, and eating disorders in children and adolescents. To avoid conceptual discrepancy, we eliminated data on stress and behavioral disorders, leaving only prevalences of anxiety and depression. LINE 99-102.

In the discussion section on LINE 614-615 we eliminated the phrase “stress management”

Point 2:

Additionally, I have concerns regarding the validation of your questionnaire, as mentioned in your third response. The study you have cited for validation purposes is not available in English, which unfortunately limits my ability to thoroughly assess the validity of your claims regarding the questionnaire's efficacy. This is a critical aspect of your research that requires further elucidation.

Response 2

The eating habits and physical activity questionnaire was initially developed and validated for adolescents in the State of Jalisco, Mexico. We attach the summary of the study by Flores et al (published in Spanish). Subsequently, the questionnaire was validated in adolescents from Mexico City (Martínez Coronado et al, published in English in Healthcare).

  1. Flores Vázquez, A.; Macedo Ojeda, G. Validation of a self-completed eating habits questionnaire for adolescents in Jalisco, Mexico.Rev Esp Nutr Comunitaria 2016, 22, 26–31.

VALIDATION OF A QUESTIONNAIRE SELF-COMPLETED

OF FOOD HABITS FOR ADOLESCENTS IN JALISCO, MEXICO

Abstract

Background: The evaluation of the quality of feeding in adolescence is essential. There are different methods, however, are most useful in clinical practice and use in populations is complicated. The objective of this study was to validate a food habits self-completion questionnaire for adolescents, through its application in a population of Jalisco, Mexico.

Methods: The questionnaire was self-completed twice for 64 high school adolescents. Exploratory factor analysis was performed; interna! consistency was assessed with Cronbach's Alpha coefficient; the correlation between test-retest scores by Pearson coefficient and intraclass correlation coefficient was measured; the correlation between the clas­sifications of the test-retest habits was assessed with Spearman and Kappa coefficient.

Results: AII variables presented significant factor loadings in their respective components. Interna! consistency was good for most sections. Pearson correlations were obtained fair and ICC values were mostly almost perfect. Spearman correlation was moderate and kappa coefficient was substantial.

Conclusions: Validation of the questionnaire was satisfactory. lts use in adolescents will allow obtain valid results and evaluate large populations without investing excessive human and time resources.

Key words: Questionnaire. Validation study. Food habits. Adolescents.

  1. Martínez Coronado, A.; Lazarevich, I.; Gutiérrez Tolentino, R.; Mejía Arias, M.Á.; Leija Alva, G.; Radilla Vázquez, C.C. Construct Validity of a Questionnaire on Eating and Physical Activity Habits for Adolescents in Mexico City. Healthcare 2023, 11, 2314. https://doi.org/10.3390/healthcare11162314

Point 3:

Another point of concern is your use of participants who are under 12 years of age. You have noted that the questionnaire has been validated for this age group, but given the aforementioned issue with the validation study's accessibility, this claim remains unsubstantiated. It is imperative that this issue be addressed to ensure the reliability and ethical compliance of your research.

Response 3

Flores' initial questionnaire was applied and validated for adolescents between 12 and 15 years of age.

The questionnaire validated in Mexico City was applied and validated in adolescents between 11 and 12 years of age. We attach a summary (Martínez Coronado, A.; Lazarevich, I.; Gutiérrez Tolentino, R.; Mejía Arias, M.Á.; Leija Alva, G.; Radilla Vázquez, C.C. Construct Validity of a Questionnaire on Eating and Physical Activity Habits for Adolescents in Mexico City. Healthcare 2023, 11, 2314. https://doi.org/10.3390/healthcare11162314

Abstract: The assessment of eating and physical activity habits is an important step in promoting healthy behaviors among the adolescent population and is key in the prevention and management of chronic non-communicable diseases, such as obesity, diabetes, and cardiovascular disease. For this purpose, reliable and valid measuring instruments are essential. In this context, the aim of this article is to present the validation of a self-report questionnaire on eating and physical activity habits among adolescents in Mexico City. In order to validate the questionnaire, a cross-sectional study was conducted with a sample of 2710 adolescents between 11 and 12 years of age, the piloting of the questionnaire was carried out in September 2022 with a focus group, and the programming of the anthropometric measurements was established with the Federal Educational Authority of CDMX, as well as the application of the questionnaire to 33 schools, with these activities being scheduled from 7 November 2022 to 3 February 2023 and having an application duration of 15–25 min for each of the groups to which it was applied; the questionnaire that was applied consists of 31 questions that refer to the frequencies, quantity, or performance of behaviors related to the frequency and type of food, type of physical activity and behaviors related to the act of eating referring to the place where it is carried out (home or away from home) and with whom it is carried out (alone or in company), and about the individual’s lifestyle. Subsequently, the reliability of the instrument was evaluated using Cronbach’s alpha coefficient, and an exploratory factor analysis was conducted to determine the structure of the questionnaire. The results obtained showed that the questionnaire was adequately reliable (α = 0.778) with an eight-factor structure: four questions on mealtime frequencies, four questions on physical activity and lifestyles, six questions on the consumption of high-calorie foods, four questions on company and food consumption, four questions on the consumption of vegetables and fruits, four questions on the place of food consumption, two questions on the consumption of alcoholic beverages, and three questions on the consumption of sugary drinks, plain water, and milk. In conclusion, the self-report questionnaire on eating and physical activity habits among adolescents in Mexico City is reliable, has adequate internal consistency, and can therefore be used as a useful tool for the evaluation of eating and physical activity habits in this population.

Point 4:

In light of these issues, I recommend another round of revision. It is essential that these concerns are adequately addressed to meet the publication's standards. I appreciate your dedication to this research and look forward to seeing the revised manuscript.

Response 4

We thank the reviewer for the valuable observations and each of them were addressed and undoubtedly strengthened the final document.

Reviewer 2 Report

Comments and Suggestions for Authors

Despite the efforts of authors to incorporate changes to produce a stronger version, I regret to inform them that the quality of the final version submitted is still of very low scientific quality to produce a decent, meaningful contribution to the scientific literature. 

Comments on the Quality of English Language

Not Applicable

Author Response

Response to Reviewer 2.

Comments and Suggestions for Authors

Despite the efforts of authors to incorporate changes to produce a stronger version, I regret to inform them that the quality of the final version submitted is still of very low scientific quality to produce a decent, meaningful contribution to the scientific literature.

We thank the reviewer for the observations made to the manuscript that allowed it to be improved.

Reviewer 3 Report

Comments and Suggestions for Authors

The work has improved. congratulations.

Author Response

Response to Reviewer 3.

Comments and Suggestions for Authors

The work has improved. congratulations.

We thank the reviewer for the observations made to the manuscript that allowed it to be improved.

Reviewer 4 Report

Comments and Suggestions for Authors

The author responses were sufficient.

Author Response

Response to Reviewer 4.

Comments and Suggestions for Authors

The author responses were sufficient.

We thank the reviewer for the observations made to the manuscript that allowed it to be improved.
